# Application of Three-Repetition Tests Scheme to Improve Integrated Circuits Test Quality to Near-Zero Defect

**DOI:** 10.3390/s22114158

**Published:** 2022-05-30

**Authors:** Chung-Huang Yeh, Jwu-E Chen

**Affiliations:** Department of Electrical Engineering, National Central University, Taoyuan 300, Taiwan; jwu.e.chen@gmail.com

**Keywords:** guardband test, test errors, test specification, defect level, test quality

## Abstract

In this research, the normal distribution is assumed to be the product characteristic, and the DITM (Digital Integrated Circuit Test Model) model is used to evaluate the integrated circuits (IC) test yield and test quality. Testing technology lags far behind manufacturing technology due to the different rates of development of the two technologies. As a result, quality control will pose significant challenges in pursuing high-quality near-zero defect products (automotive and biomedical electronics and avionics, etc.). In order to ensure product quality, we propose an effective repeated testing method (three-repetition tests scheme, TRTS), which utilizes the move test guardband (TGB) to improve the test yield and test quality. Based on the data in the International Roadmap for Devices and Systems table in 2021, the DITM model is used to estimate the future trend of semiconductor chip test yield, and the retest method (TRTS) is applied improve the test results. The method of repeated testing can increase the test yield and increase the shipment of semiconductor products. By estimating the test cost and profit, the method of repeated testing can obtain chips with near-zero defects with more corporate profits through increased product shipments.

## 1. Introduction

As automotive electronic equipment becomes more widely used, the number of electronic components in automobiles continues to rise. Consequently, with such a large number of electronic components, quality and safety have become critical issues. Semiconductor manufacturers have begun to invest heavily in improving semiconductor production equipment, and several effective testing solutions have been proposed to prevent defective electronic products from reaching the market. For example, the American Automotive Electronics Council AEC adopts a part average testing (part average testing, PAT) method AEC-Q001 [1,2] to eliminate abnormal parts from the total parts and increase product reliability. Furthermore, semiconductor integrated circuits (IC) test manufacturers employ the retest method [3,4,5,6,7,8,9,10,11] in the production process to improve the test yield (Y_t_). For example, Teslence Technology Co., Ltd. (TT) has developed a new testing method [3] that has been applied to the test production line of the test factory ASE Technology Holding Co. (The world’s largest chip packaging and testing services), Ltd., to improve the test yield (Y_t_) of chip products. Of course, academia has also proposed many effective methods to solve the test problem on the semiconductor production line.

For example, Ken Chau-Cheung Cheng et al. [4]. analyzed the characteristics of test-induced defect patterns and defined the characteristics that machine learning algorithms can use to automatically detect induced defects. Therefore, defective dies caused by wafer testing can be retested to improve yield. Furthermore, S.C. Horng et al. [5]. proposed a two-stage approach based on ordinal optimization theory to solve the problem of obtaining a good enough threshold and achieving less overkill and retesting in reasonable computation time. Moreover, Hardi Selg et al. [6]. proposed a way to apply machine learning to efficiently predict retests. Supervised learning on a predefined subset of wafers includes information about first and retest runs. Experimental results on real product data demonstrate the effectiveness of the retest success prediction method, resulting in a significant optimization of manufacturing test time. The methods proposed above reduce test time and cost while increasing test yield and product quality.

This paper introduces a concept of probability, assumes normal distribution for product characteristics, quantifies the manufacturing process and test, and applies the test quality and yield model (DITM) to analyze IC characteristics and evaluate test yield and test quality. Chips produced in the future will become increasingly complex, making it more difficult and time-consuming to determine whether the chip is good or bad. Furthermore, semiconductor technology is rapidly evolving, and testing technology lags far behind design and manufacturing [12,13,14]. Using relatively poor testers to measure high-quality chips has become a major issue. Therefore, we propose repeating the test method (three-repetition tests scheme, TRTS) in pursuit of high-quality production methods. Using the ATE (automated test equipment) with poor performance, changing the test method, and moving the test guardband (TGB) [15,16,17,18,19,20,21], repeatedly find a truly zero-defect and reliable product to ensure that the chip can function normally and consistently. We referred to the IRDS (International Roadmap for Devices and Systems 2021) data [22] sheet to estimate the future test yield distribution. Using estimated results, we showed that repeated testing (TRTS) could improve test yield and quality. Furthermore, it provides more secure automotive and avionics chips and has the potential to increase total chip shipments. High-quality chips (near-zero defects, i.e., 10 ppm) provide more stable and safer performance, and the price of chips is several times the price of a batch of chips. By improving test yield and quality, the company’s total profit can be increased, and the capacity shortage caused by the COVID-19 epidemic can be alleviated.

## 2. Errors in the Semiconductor Chip Manufacturing Process and Test Process

Temperature and exposure errors, as well as chemical-concentration errors during etching, are instances of manufacturing errors [12,13,14]. Apart from environmental factors during manufacturing, the accuracy of the ATE (IC tester) during subsequent testing has an impact on test yield and test quality. After the VLSI manufacturing is completed, it will be judged as a good product if the functions and parameters meet the design specification (DS). Otherwise, it will be judged as a defective product. Suppose the total number of chips to be produced is N, G is expressed as the number of good chips, and Y_m_ = G/N can be used to express the manufacturing yield Y_m_ or true yield, as shown in Figure 1.

We judge whether the chip passed or failed the VLSI test (as shown in Figure 1) by setting the test specification TS sent by the ATE (IC tester). If the function and parameters meet the test specifications (part of P), the test is deemed to pass. Likewise, functions and parameters that do not meet the test specification (part of F) are rejected. The test yield (Y_t_) can be expressed as Y_t_ = P/N. There will be no yield loss if the testing process is perfect. The actual chip test cannot be perfect, and the inaccuracy of the operation or tester during the test may result in a killing error (α) and missing error (β). Killing errors (α) are defined as the number of good chips that fail the test [23], which will increase production cost and yield loss. Bad chips that pass the test are defined as missing errors (β), which will lead to quality problems, and customers will reject products.

### 2.1. Estimation and Calculation of the Formula for Manufacturing Yield (Y_m_)

The normal distribution is often used as a reference in statistical analysis because of its accuracy in data studies (Figure 1), which is well suited as a hypothetical distribution for parameters in a study. The characteristic parameters of the components after manufacturing show an uncertain distribution due to uncertain factors in the wafer manufacturing process, such as etching, chemical doping, and personnel operation problems. In this research, we assume that the DUT (Device under test) chip delay time is normal, i.e., Chip (x) = N(x; μ_M_, σ_M_) with mean μ_M_ and standard deviation σ_M_ [24,25]. The manufacturing yield (Y_m_) is the probability of the area under the normal curve between the coordinates x = DS and x = −∞, i.e., P[−∞ < X < DS]. Thus, we find as follows:(1)$$\begin{array}{l} \mathrm{Manufacturing}\;\mathrm{Yield}\left(\%\right)=Y_{m}\\ =\int_{-\infty }^{\mathrm{DS}}\mathrm{Chip}\left(x\right)\mathrm{dx}\\ =\int_{-\infty }^{\mathrm{DS}}\frac{1}{\sqrt{2\pi }\sigma_{M}}e^{-\frac{1}{2}{\left(\frac{X-\mu_{M}}{\sigma_{M}}\right)}^{2}}\;\mathrm{dx}\\ =\int_{-\infty }^{\frac{\mathrm{DS}-\mu_{M}}{\sigma_{M}}}\frac{1}{\sqrt{2\pi }}e^{-\frac{1}{2}{\left(x\right)}^{2}}\mathrm{dx} \end{array}$$

Assume the DS is 0.858 GHz (DS = 1165 ps), the average of the normal distribution μ_M_ = 1000 ps (picosecond), and the standard deviation σ_M_ = 100 ps for a circuit. Figure 2 depicts the chip delay time distribution X ~ N (x; μ_M_ = 1000 ps and σ_M_ = 100 ps). The horizontal axis is the time parameter of the circuit characteristics, and the vertical axis is the probability density of the time parameter. The chips manufactured by the foundry are classified according to the DS, and they can be divided into good products (Good) that meet the design specifications and bad products (Bad) that do not meet the design specifications. Manufacturing yield (true yield) can be derived as Y_m_ = P[Good] = P[X < DS] = 95% using the formula (1).
$$Y_{m}\left(\%\right)=\int_{-\infty }^{1165}\frac{1}{100\sqrt{2\pi }}e^{\frac{-{\left(x-1000\right)}^{2}}{2\times_{100}{}^{2}}}\mathrm{dx}=95\%$$

### 2.2. Estimation of the Formula for Test Yield (Y_m_)

Figure 3 presents the production process in the semiconductor industry, which is mainly composed of wafer manufacturing, wafer testing, chip packaging, and post package testing. Chip testing is divided into two stages: CP (Chip Probing) testing, which is a wafer test (Wafer testing), and FT (Final Test) testing, which is performed after the chip has been packaged to save on packaging costs. The CP test requires a probe card, whereas the FT test uses a load board and socket. The IC parameter test verifies the DUT using electrical detection and determines whether the DUT is qualified by measuring the DUT output signal. Many projects are included in IC testing, such as Delay Test, Parameter Test, and Function Test. Furthermore, three general categories of defect classes are as follows: (A) bridge, (B) open circuit, and (C) parametric defects. This paper compares the timing of the chip to determine whether it is good or bad (the strobe timing of the tester and the chip delay time of the chip).

A fixed reference voltage (Vref) is supplied to one input of a comparator [21], and the output voltage value determines the result (If x1 < Vref, then V0 = 0; If x1 > Vref, then V0 = Vcc) in the ATE (IC tester) system model, as shown in Figure 4. The tester sends the strobe signal to compare the timing with the product response in an ATE. Hence, D-type flip-flops determine whether the product passes or fails based on the output of the timing comparator. The tester (ATE) compares the results of two signal timings to determine whether the DUT is a “pass” or a “fail” (Figure 4) in this study. If the chip delay time arrives faster than the signal sent by the tester (X1 < X2), a pass signal is sent, and the chip is regarded as a good chip (G). Conversely, the chip delay time arrives slower than the signal from the tester (X1 > X2), and the tester (ATE) sends a fail signal, treating the chip as a bad one (B).

The added value of testing to the product is mainly to improve the quality. Uncertain semiconductor process errors have resulted in partially defective products during manufacturing, but testing can improve this state. Due to ATE inaccuracy, the signal ST from the tester has edge placement during the testing process (Automatic test equipment). The performance N of a testing device’s capability (Tester) is assumed to be normal. We found X ~ N (x; μ_T_, σ_T_), mean μ_T,_ and standard deviation σ_T_ using the tester. The test yield Y_t_ is calculated as Y_t_ = P[pass] = P[X < Y] and expressed as follows:(2)$$\begin{array}{l} \mathrm{Test}\;\mathrm{Yield}\;\left(Y_{t}\right)=\;P\left[x<y\right]=\;P\left[\mathrm{pass}\right]\\ =\int_{-\infty }^{\infty }\mathrm{Chip}\left(x\right)\int_{x}^{\infty }\mathrm{Tester}\left(y\right)\mathrm{dydx}\\ =\int_{-\infty }^{\infty }\frac{1}{\sqrt{2\pi }\sigma_{M}}e^{-\frac{1}{2}{\left(\frac{X-\mu_{M}}{\sigma_{M}}\right)}^{2}}\int_{x}^{\infty }\frac{1}{\sqrt{2\pi }\sigma_{T}}e^{-\frac{1}{2}{\left(\frac{y-\mu_{T}}{\sigma_{T}}\right)}^{2}}\mathrm{dydx}\;\\ =\int_{-\infty }^{\infty }\frac{1}{\sqrt{2\pi }}e^{-\frac{1}{2}{\left(x\right)}^{2}}\int_{\frac{\mu_{M}+\sigma_{M}x-\mu_{T}}{\sigma_{T}}}^{\infty }\frac{1}{\sqrt{2\pi }}e^{-\frac{1}{2}y^{2}}\mathrm{dydx} \end{array}$$

In the measurement of product quality, the defect level (DL) represents the quality of the product, which can be expressed as DL = P[Bad | Pass] = P[(X > DS) ∩ (X < ST)]/P[X < ST]. The unit of DL is ppm.
(3)$$\begin{array}{l} \mathrm{Missing}\;\mathrm{Errors}\;=\int_{\mathrm{DS}}^{\infty }\mathrm{Chip}\left(x\right)\int_{x}^{\infty }\mathrm{Tester}\left(y\right)\mathrm{dydx}\\ =\int_{\mathrm{DS}}^{\infty }\frac{1}{\sqrt{2\pi }\sigma_{M}}e^{-\frac{1}{2}{\left(\frac{X-\mu_{M}}{\sigma_{M}}\right)}^{2}}\int_{x}^{\infty }\frac{1}{\sqrt{2\pi }\sigma_{T}}e^{-\frac{1}{2}{\left(\frac{y-\mu_{T}}{\sigma_{T}}\right)}^{2}}\mathrm{dydx}\\ =\int_{\frac{\mathrm{DS}-\mu_{M}}{\sigma_{M}}}^{\infty }\frac{1}{\sqrt{2\pi }}e^{-\frac{1}{2}{\left(x\right)}^{2}}\int_{\frac{\mu_{M}+\sigma_{M}x-\mu_{T}}{\sigma_{T}}}^{\infty }\begin{array}{c} \frac{1}{\sqrt{2\pi }}e^{-\frac{1}{2}y^{2}}\mathrm{dydx} \end{array} \end{array}$$
(4)$$\begin{array}{l} \mathrm{DL}\;\left(\mathrm{Defect}\;\mathrm{Level}\right)=\frac{P\left[\mathrm{Bad}|\mathrm{Pass}\right]}{Y_{t}}\\ =\frac{\mathrm{Missing}\;\mathrm{Errors}}{Y_{t}}\\ =\frac{\int_{\mathrm{DS}}^{\infty }\frac{1}{\sqrt{2\pi }\sigma_{M}}e^{-\frac{1}{2}{\left(\frac{X-\mu_{M}}{\sigma_{M}}\right)}^{2}}\int_{x}^{\infty }\frac{1}{\sqrt{2\pi }\sigma_{T}}e^{-\frac{1}{2}{\left(\frac{y-\mu_{T}}{\sigma_{T}}\right)}^{2}}\mathrm{dydx}}{Y_{t}}\\ =\frac{\int_{\frac{\mathrm{DS}-\mu_{M}}{\sigma_{M}}}^{\infty }\frac{1}{\sqrt{2\pi }}e^{-\frac{1}{2}{\left(x\right)}^{2}}\int_{\frac{\mu_{M}+\sigma_{M}x-\mu_{T}}{\sigma_{T}}}^{\infty }\frac{1}{\sqrt{2\pi }}e^{-\frac{1}{2}y^{2}}\mathrm{dydx}}{Y_{t}} \end{array}$$

Let $R_{1t}^{1p}$ represents the traditional test method for testing the DUT (testing the DUT only once).

Taking 100 ppm as an example, it means that in one million products that are produced, there may be 100 defective products. A higher defect rate (1000 ppm) indicates a higher percentage of return rate; a lower defect degree (10 ppm) indicates a lower percetage of return rate. This parameter is usually used to as an indicator of product quality. Each product has different quality requirements. Generally speaking, taking general consumer electronic products as an example, DL = 200~300 ppm (part per million) should be a quality acceptable to both manufacturers and consumers. However, in the field of biomedical electronics or automotive electronics, chip manufacturers must implement more stringent quality control in their chip manufacturing products and must meet higher safety quality requirements. For many fairly important safety functions currently controlled by semiconductors, such as high reliability critical electronic systems, the minimum requirement of quality is 10 ppm. Especially in some critical and important parts, manufacturers will even increase the accuracy of measurement by replacing the unit of defect rate from the commonly used Parts Per Million (ppm) to one billionth (Parts Per Billion, ppb). We all know that the pursuit of zero-defect products is the goal of the semiconductor industry. By reducing the defect rate of the chip, the malfunction of electronic parts can be reduced, and the stability of the chip can be improved. Of course, a lower defect rate implies a higher price tag and better profits for the company.

## 3. IC Test Yield and Product Quality Are Strongly Affected by the ATE Accuracy

OTA (Overall Timing Accuracy) is the accuracy specification of the ATE (Semiconductor test equipment, IC tester). The test standard deviation σ_T_ is set to one-third of OTA (Overall Time Accuracy). The OTA can be expressed as OTA = 3 × σ_T_. The smaller the OTA value is, the higher the tester’s accuracy. In contrast, the larger the OTA value is, the lower the tester’s accuracy is. The ideal IC tester OTA value needs to be increased simultaneously with the integrated circuit technology to ensure the test yield and quality. Here we defined r as the testing and manufacturing accuracy ratio; thus, r = OTA/3σ_M_. The smaller the ratio, the higher the accuracy of the ATE, implying that the testing capability is better than the manufacturing capability [26,27]. For example, DUT characteristics X ~ N (x; μ_M_ = 1000 ps and σ_M_ = 100 ps). According to the following calculation, we can get a 95% manufacturing yield Y_m_. 

In Figure 5 and Table 1, if we have chosen a less accurate tester with σ_T_ of 60 ps (the larger the OTA, the lower the accuracy). Thus, tester accuracy became OTA = 180 ps (OTA = 3 × σ_T_ = 180 ps); the testing and manufacturing accuracy ratio was r = 0.6 (r = OTA/3σ_M_ = 180/300 = 0.6). DL was set to 300 ppm, and a test yield Y_t_ of 59.5% ($R_{1t}^{1p}$) was obtained using TS = 1028 ps to test DUT.

Next, we had selected a high accuracy tester to test the DUT (the smaller r, the higher the tester’s accuracy), the tester’s characteristic parameter σ_T_ is 30 ps, and if the tester’s accuracy OTA is 90 ps (OTA = 3σ_T_ = 3 ×30=90 ps), then r = 0.3 (r = 90/300 = 0.3). If μ_T_ = 1107 ps (TS = 1107 ps), the quality requirement is set at 300 ppm, where Y_t_ (Test Yield) = 84.7%. The above simulation analysis shows that using a high-precision (OTA) ATE improves test quality and increases test yield by approximately 25.2% (Y_t (OTA_
_= 30 ps__)_− Y_t_
_(OTA_
_= 90 ps__)_ = 84.7% − 59.5% = 25.2%). The test yield was increased for testers with high accuracy (OTA = 90 ps), and test quality was also maintained. From the above example, we can understand that choosing a low-accuracy tester (OTA) will result in more missing errors and killing errors, resulting in a poor test yield. Conversely, renting a high-precision (OTA) tester can result in a high test yield, and the overall profit will decrease due to the high cost of renting a tester. Therefore, test decision-makers should consider the cost of the tester and the test yield when selecting an ATE (IC tester) that meets the cost requirements of the market and consumers.

Of course, some consumer products do not need high-quality requirements. Take the AMD CPU of a desktop PC as an example, DL = 300 ppm (part per million) should be acceptable to the manufacturer. However, in some products, such as automotive biomedicine electronics, one needs high standard quality close to zero defects (10 ppm) [28,29,30,31,32]. For example, after being manufactured in the same conditions by a foundry, limiting the DL to 10 ppm (Figure 6), the TGB is moved and tested using traditional test methods $R_{1t}^{1p}$ [33,34]. When using the test specification TS = 1073 ps, the test yield dropped to 75.8%, according to the previous calculation to estimate the test yield (OTA = 90 ps). Despite the loss of 8.9% of the test yield, it is exchanged for stable and high-quality chips (Figure 6). High-quality products represent higher sales prices and can bring the company a better market reputation and popularity.

Test quality at DL = 300 ppm with the test guardband to the right (TGB) (the smaller the test guardband is set) shows that test yield is better but has poor test quality. In contrast, test quality at DL = 10 ppm with the test guardband to the left (the larger the test guardband setting is) shows that test yield is poor but has better test quality. The larger the guardband, the higher the rate of killing error, but the quality of the shipment can be guaranteed. Product test yield and quality are interchangeable when using traditional test methods and moving the test guardband. Uncertain factors in the semiconductor manufacturing process will inevitably lead to product defects in some cases. As a result, the back-end testing process must correctly set the test guardband to shave the defective chips and improve product quality effectively.

## 4. A New Scheme for Three-Repetition Tests

Semiconductor testing has faced many challenges in recent years. The chip clock speed is getting faster and faster, the pin count will reach thousands of pins, and the electrical power consumption is getting higher and higher. According to the ITRS roadmap, test and manufacturing technologies evolve at different speeds [12,13,14], and the development speed of future testers will lag behind semiconductor process technologies. Referring to Moore’s Law, integrated circuits are improving by 30% per year, but the overall rate of progress in the automatic test equipment (ATE) industry has stagnated. We know that the less accurate the ATE, the harder it is to determine if the device under test (DUT) can operate at high speeds. This uncertainty increases as the DUT speed increases. Conversely, if backward integrated circuits (IC) testers (ATE) are used to differentiate the quality of advanced chips, yield and testing quality may not be maintained at a certain level. The future looks bleak for semiconductor manufacturers and testing labs. Therefore, the semiconductor industry and academia have proposed various test and verification methods. For example, Betsy S. Greenberg [7] introduced the concept of early retesting, and they developed a model that calculated the expected return of n retests. Therefore, use these estimates to design test plans and compare their effectiveness.

Retesting has been widely used in the semiconductor IC testing industry in recent years to improve test results [3,4,5,6,7,8,9,10,11]. For example, the test method for full-speed testing [8] is to operate the chip at a clock frequency (on-chip clock controller) higher than that of the ATE to detect delay failures on the wafer. This method is undoubtedly effective when using full-speed testing. However, the test method of full-speed testing may require more up-front design work because the test circuit must be embedded in the chip. Furthermore, Kirmse et al. [9]. proposed three different wafer retest models to quickly analyze wafers and use the retest chip method to improve the test yield. This method can significantly improve the speed and efficiency of the inspection process and quickly detect faulty wafers.

Moreover, Teslence Technology Co., Ltd. (TT), the world’s largest wafer test factory, developed a new test method [3] with ASE Technology Holding Co., Ltd. (Figure 7). Here, an automatically calculated re-probing path is provided to minimize re-screening testing time. This approach maximizes product recovery while allowing for production flexibility without downtime. The test method is applied to the chip test production line, and the results show that the retest method can effectively improve the test yield.

Hence, in order to ensure product quality, we propose an effective repeated testing method (three-repetition tests scheme, TRTS), which utilizes the move test guardband (TGB) to improve the test yield and test quality. We changed the test conditions and methods and extended the test time to improve product quality and test yield while maintaining a reasonable test cost.

Figure 7 depicts a schematic of the proposed process. From the initial test processing, all tested chips are partitioned into the pass (P, DUT that has passed the first test) and fail (F, DUT that failed the first test) part. Furthermore, we chose to pass the good part (P) of the first test and retest at the same test specification. The good part (P) passes the test twice, and we call it “Repetition Tests Scheme,” and the symbol is denoted $R_{2t}^{2p}$. The test result formula of repeat test ($R_{2t}^{2p}$) is defined as follows:(5)$$\begin{array}{l} \mathrm{Test}\;\mathrm{Yield}\left(\%\right)Y_{t}\;(R_{2t}^{2p})\;\\ =\int_{-\infty }^{\infty }\mathrm{Chip}\left(X\right)\int_{x}^{\infty }\mathrm{Tester}\left(y,{\;\mu }_{T}\right)\mathrm{dy}\int_{x}^{\infty }\mathrm{Tester}\left(z,{\;\mu }_{T}\right)\mathrm{dzdx}\\ =\int_{-\infty }^{\infty }\frac{1}{\sigma_{M}\sqrt{2\pi }}e^{\frac{-{\left(x-\mu_{M}\right)}^{2}}{2\sigma_{M}{}^{2}}}\int_{x}^{\infty }\frac{1}{\sigma_{T}\sqrt{2\pi }}e^{\frac{-{\left(y-\mu_{T}\right)}^{2}}{2\sigma_{T}{}^{2}}}\mathrm{dy}\int_{x}^{\infty }\frac{1}{\sigma_{T}\sqrt{2\pi }}e^{\frac{-{\left(z-\mu_{T}\right)}^{2}}{2\sigma_{T}{}^{2}}}\mathrm{dzdx}\\ =\int_{-\infty }^{\infty }\frac{1}{\sqrt{2\pi }}e^{-\frac{1}{2}{\left(x\right)}^{2}}\int_{\frac{\mu_{M}+\sigma_{M}x-\mu_{T}}{\sigma_{T}}}^{\infty }\frac{1}{\sqrt{2\pi }}e^{-\frac{1}{2}y^{2}}dy\int_{\frac{\mu_{M}+\sigma_{M}x-\mu_{T}}{\sigma_{T}}}^{\infty }\frac{1}{\sqrt{2\pi }}e^{-\frac{1}{2}z^{2}}\mathrm{dzdx} \end{array}$$
(6)$$\begin{array}{l} \mathrm{DL}\;\left(\mathrm{Defect}\;\mathrm{Level}\right)=\frac{P\left[\mathrm{Bad}|\mathrm{Pass}\right]}{Y_{t}}\\ =\frac{\mathrm{Missing}\;\mathrm{Errors}}{Y_{t}}\\ =\frac{\int_{\mathrm{DS}}^{\infty }\frac{1}{\sigma_{M}\sqrt{2\pi }}e^{\frac{-{\left(x-\mu_{M}\right)}^{2}}{2\sigma_{M}{}^{2}}}\int_{x}^{\infty }\frac{1}{\sigma_{T}\sqrt{2\pi }}e^{\frac{-{\left(y-\mu_{T}\right)}^{2}}{2\sigma_{T}{}^{2}}}\mathrm{dy}\int_{x}^{\infty }\frac{1}{\sigma_{T}\sqrt{2\pi }}e^{\frac{-{\left(z-\mu_{T}\right)}^{2}}{2\sigma_{T}{}^{2}}}\mathrm{dzdx}}{Y_{t}}\\ =\frac{\int_{\frac{\mathrm{DS}-\mu_{M}}{\sigma_{M}}}^{\infty }\frac{1}{\sqrt{2\pi }}e^{-\frac{1}{2}{\left(x\right)}^{2}}\int_{\frac{\mu_{M}+\sigma_{M}x-\mu_{T}}{\sigma_{T}}}^{\infty }\frac{1}{\sqrt{2\pi }}e^{-\frac{1}{2}y^{2}}\mathrm{dy}\int_{\frac{\mu_{M}+\sigma_{M}x-\mu_{T}}{\sigma_{T}}}^{\infty }\frac{1}{\sqrt{2\pi }}e^{-\frac{1}{2}z^{2}}\mathrm{dzdx}}{Y_{t}} \end{array}$$

The increase in the number of repeated tests affects the test results. If we pass the test once (the good part (P), retest n times (Figure 8) with the same test specification (where “n” represents the number of additional tests for the passing (P) DUT), the formula for multiple tests ($R_{\mathrm{nt}}^{\mathrm{np}}$) is defined as follows:(7)$$\begin{array}{l} \mathrm{Test}\;\mathrm{Yield}\left(\%\right)Y_{t}=(R_{\mathrm{nt}}^{\mathrm{np}})Y_{\mathrm{pp}\ldots }\\ =\int_{-\infty }^{\infty }\mathrm{Chip}\left(X\right)\int_{x}^{\infty }\mathrm{Tester}\left(y,{\;\mu }_{T}\right)\mathrm{dy}\int_{x}^{\infty }\mathrm{Tester}\left(z,{\;\mu }_{T}\right)\mathrm{dz}\ldots \ldots \ldots \ldots .\int_{x}^{\infty }\mathrm{Tester}\left(w,{\;\mu }_{T}\right)\mathrm{dwdx}\\ =\int_{-\infty }^{\infty }\frac{1}{\sigma_{M}\sqrt{2\pi }}e^{\frac{-{\left(x-\mu_{M}\right)}^{2}}{2\sigma_{M}{}^{2}}}\int_{x}^{\infty }\frac{1}{\sigma_{T}\sqrt{2\pi }}e^{\frac{-{\left(y-\mu_{T}\right)}^{2}}{2\sigma_{T}{}^{2}}}\mathrm{dy}\int_{x}^{\infty }\frac{1}{\sigma_{T}\sqrt{2\pi }}e^{\frac{-{\left(z-\mu_{T}\right)}^{2}}{2\sigma_{T}{}^{2}}}\mathrm{dz}\ldots .\int_{x}^{\infty }\frac{1}{\sigma_{M}\sqrt{2\pi }}e^{\frac{-{\left(w-\mu_{M}\right)}^{2}}{2\sigma_{M}{}^{2}}}\mathrm{dwdx}\\ =\int_{-\infty }^{\infty }\frac{1}{\sqrt{2\pi }}e^{-\frac{1}{2}{\left(x\right)}^{2}}\int_{\frac{\mu_{M}+\sigma_{M}x-\mu_{T}}{\sigma_{T}}}^{\infty }\frac{1}{\sqrt{2\pi }}e^{-\frac{1}{2}y^{2}}\mathrm{dy}\int_{\frac{\mu_{M}+\sigma_{M}x-\mu_{T}}{\sigma_{T}}}^{\infty }\frac{1}{\sqrt{2\pi }}e^{-\frac{1}{2}z^{2}}\mathrm{dz}\ldots \ldots \int_{\frac{\mu_{M}+\sigma_{M}x-\mu_{T}}{\sigma_{T}}}^{\infty }\frac{1}{\sqrt{2\pi }}e^{-\frac{1}{2}w^{2}}\mathrm{dwdx} \end{array}$$
(8)$$\begin{array}{l} \mathrm{DL}\;\left(\mathrm{Defect}\;\mathrm{Level}\right)=\frac{P\left[\mathrm{Bad}|\mathrm{Pass}\right]}{Y_{t}}\\ =\frac{\mathrm{Missing}\;\mathrm{Errors}}{Y_{t}}\\ =\frac{\int_{\mathrm{DS}}^{\infty }\mathrm{Chip}\left(X\right)\int_{x}^{\infty }\mathrm{Tester}\left(y,{\;\mu }_{T}\right)\mathrm{dy}\int_{x}^{\infty }\mathrm{Tester}\left(z,{\;\mu }_{T}\right)\mathrm{dz}\ldots \ldots \ldots \ldots .\int_{x}^{\infty }\mathrm{Tester}\left(w,{\;\mu }_{T}\right)\mathrm{dwdx}}{Y_{t}}\\ =\frac{\int_{\frac{\mathrm{DS}-\mu_{M}}{\sigma_{M}}}^{\infty }\frac{1}{\sqrt{2\pi }}e^{-\frac{1}{2}{\left(x\right)}^{2}}\int_{\frac{\mu_{M}+\sigma_{M}x-\mu_{T}}{\sigma_{T}}}^{\infty }\frac{1}{\sqrt{2\pi }}e^{-\frac{1}{2}y^{2}}\mathrm{dy}\int_{\frac{\mu_{M}+\sigma_{M}x-\mu_{T}}{\sigma_{T}}}^{\infty }\frac{1}{\sqrt{2\pi }}e^{-\frac{1}{2}z^{2}}\mathrm{dz}\ldots \ldots \int_{\frac{\mu_{M}+\sigma_{M}x-\mu_{T}}{\sigma_{T}}}^{\infty }\frac{1}{\sqrt{2\pi }}e^{-\frac{1}{2}w^{2}}\mathrm{dwdx}}{Y_{t}} \end{array}$$

### Selecting the Number of Valid Retests Determines the Company’s Profitability

The main purpose of integrated circuit testing (IC testing) is to confirm if the chip meets the design specifications, and to ensure the quality and reliability of semiconductor products. In recent years, with the rapid development of integrated circuits, the functions of chips have become more and more powerful and their speed has become faster and faster. With such a large-scale circuit, test verification becomes more and more difficult. Therefore, how to find a fast and effective test method has become a very important issue. In particular, in order to ensure the reliability of critical electronic products, strict quality control is required to eliminate all defective parts in the total number of parts. However, as advances in integrated circuit tester (ATE) technology have lagged behind the pace of development in semiconductor manufacturing, ATE advancements have stalled, resulting in increasingly poor test results(low test yield). Therefore, it will be a great challenge to use an ATE with performance lagging behind the process capability to pick out highly reliable electronic products. Repeat testing (repetition tests scheme) can improve test yield and quality; however, test costs will gradually increase with increasing test time and the number of tests. Moreover, when the increased test cost is higher than the increased profit (increased test yield), there is no point repeating the test. Blind retesting wastes manpower and increases testing costs. Furthermore, when implementing the test-retest approach, the cost of testing must be considered. Therefore, choosing the most effective and optimal number of tests can avoid blind retesting and get the best profit. In order to avoid the cost problem and the waste of manpower caused by directionless testing, with the following examples, we use different test schemes ($R_{1t}^{1p},{\;R}_{2t}^{2p},{\;R}_{3t}^{3p}{\;\mathrm{and}\;R}_{4t}^{4p}$) ) to test the DUT. Through the estimated results, we compare the test yield and test cost obtained by different test schemes. Finally, through the analysis of profit and loss statistics of test yield and test cost, we get the optimized retest plan.

The chip pricing model in the international market is 8:20 [35]. When the chip manufacturing cost is 8, the market price is 20. For example, assuming the IC costs $8 to manufacture, the IC sells for $20. Testing consumes an increasing portion of the manufacturing cost in the manufacture and testing of semiconductor chips. In general, chip testing costs 5% of total manufacturing costs [12,13,14]. Assuming that company “B” produces 100 million chips and the manufacturing cost per wafer is $8, the total cost of required testing is about $40 million (100,000,000 × 8 × 5% = $40,000,000). As shown in Figure 9, the manufacturing schedule and product variation are considered, and the data in Table 2 is substituted into the DITM model (repetition tests scheme). For example, we design a chip with DS = 1165 ps (0.858 GHz) with X ~ N (x; μ_M_ = 1000 ps and σ_M_ = 100 ps), σ_M_ = 100 ps. Using the above formula (1)–(3), a manufacturing yield of Y_m_ = 95% was obtained (Figure 9 and Table 2). When the tester (OTA = 120 ps, r = 0.4) is selected to test the DUT, the test quality DL is set to 300 ppm. We can obtain Y_t_ = 77.76% if the DUT is tested using the traditional test method $R_{1t}^{1p}$ and the test specification TS = 1082 ps.

Under the same test conditions (DL = 300 ppm), repeating the test $R_{2t}^{2p}$ to test the DUT can improve the yield from Y_t_ = 77.76% ($R_{1t}^{1p}$) to Y_t_ = 83.47% ($R_{2t}^{2p}$). Thus, company “B” can sell 5,710,000 chips per year, generating an additional $114.2 million in annual revenue (Figure 10 and Table 2). (100,000,000 × 20 × 5.71% = 114.2 million). After deducting the cost of testing for two repeat tests, an additional $34.2 million (114.2 − 40 − 40 = $34.2 million) can be earned.

According to the above estimate, the DUT is tested using the repeated test $R_{3t}^{3p}$. We set the TS value (Ts = 1145 ps) to test the DUT, and the test yield ranged from Y_t_ = 77.76% ($R_{1t}^{1p}$) to Y_t_ = 85.6% ($R_{3t}^{3p}$). The test yield increased by 7.84% (85.6% − 77.76% = 7.84%). After deducting the cost of three repeat tests, the additional income became $76 million (156.8 − 40 − 40 − 40 = $36.8 million). Next, we used the repeated test $R_{4t}^{4p}$ (Ts = 1157 ps) to test the DUT, while the test yield (Y_t_) could be improved from 77.76% to 86.76%, although the test yield of four retests is slightly higher than that of three tests by 1.11% ($R_{4t}^{4p}$ − $R_{3t}^{3p}$ = 86.76 − 85.6 = 1.16%). However, after deducting the cost of testing, the total profit is lower than that of the three tests. We can understand that the cost of repeating the test $R_{4t}^{4p}$ four times is greater than the profit from increasing production (10 million × 20 × 1.16% = $23.2 million, 23.2 − 40 = −$16.8 million). The above analysis shows that the proposed repeated testing method can improve test yield; however, the more tests performed, the higher the cost. The required testing cost is higher than the profit added by the testing (the yield increased by the repetitive testing method), resulting in a decrease in the company’s overall profit when using the repetitive testing method. As a result, the test executor must select the appropriate number of tests while avoiding repeated tests (blind retesting). Furthermore, selecting an appropriate number of tests can help to avoid unnecessary retests, save manpower and time, and lower testing costs [36]. Compare the test results of the above two different test qualities (10ppm and 300ppm). We measure the increased test cost, test yield, and profit. We can clearly understand from the above results that the triple test (TRTS) testing strategy can save manpower and generate the best profit based on cost estimation and judgment. The yield and profit obtained by triple repeat testing (TRTS) is the first choice to optimize the number of retests.

## 5. The Use of the TRTS in IRDS 2021 Data

The rate of advancement in future semiconductors is unpredictable. Thus, we used the electrical characteristics of existing circuits and fabrication techniques to predict future trends in product distribution. Among them, the manufacturing schedule parameter (α) is used to estimate future product performance parameters. We assumed normal product distribution (Y_m_ = 95%) and used the previously established digital integrated-tester model (DITM) and data from the IRDS 2021. Considering the close relationship that we found between the manufacturability parameter (C_m_) and Y_m_ (the higher the C_m_, the higher the Y_m_), we have
(9)$$\mathrm{Cm}\;=\frac{\left(\mathrm{DS}-\mu_{M}\right)}{\sigma_{M}}$$
to derive
(10)$$\mathrm{DS}\;={\;\mu }_{M_{n+1}}+{\mathrm{Cm}\times \sigma }_{M_{n+1}}$$
and then
(11)$$\frac{\sigma_{M_{n+1}}}{\mu_{M_{n+1}}}={\left(\frac{\sigma_{M_{n}}}{\mu_{M_{n}}}\right)}^{\alpha }$$

For example, we applied Equations (9)–(11) to the chip data from the IRDS 2021, using DS = 303 ps (3.3 GHz) with circuit-property parameter X ~ N (x; μ_M_ = 195 ps, σ_M_ = 65 ps) for the year 2022 (Table 3), assuming Y_m_ = 95% when C_m_ = 1.65, and setting α = 1. Then 294 = μ_M+1_ +1.65 × σ_M_ and $\frac{\sigma_{M_{n+1}}}{\mu_{M_{n+1}}}={\left(\frac{65}{195}\right)}^{1}$ were substituted into the formula, and the 2023 circuit-property parameter was estimated as X ~ N (x; μ_M_ = 190 ps, σ_M_ = 63 ps).

The rapid progress of semiconductor manufacturing technology has stagnated relative to the progress of test equipment (ATE), causing tester technology to lag behind semiconductor manufacturing technology by more than a generation. Considering the ITRS roadmap data [12,13,14] and the actual operation of the test house, the OTA value set at 100 ps is a value that is in line with the current situation. Next, we applied the method of repeated testing in the 2021 IRDS table [22]. Referring to Figure 11 and Table 3, the DS of the DUT in 2021 is 313 ps, and its characteristic parameters are X ~ N(x; μ_M_ = 202 ps, σ_M_ = 67 ps). We get a 95% manufacturing yield (Y_m_) by substituting the above-estimated formula. Next, limiting the DL to 300 ppm, using a tester (ATE) with OTA as a 100 ps characteristic parameter, and using the traditional test method $R_{1t}^{1p}$ (TS = 240 ps) to test the DUT, we get Y_t_ = 69.4%. Next, referring to the IRDS 2021 datasheet, when DS = 278 ps (3.6 GHz) in 2025, the OTA value of the tester (ATE) is 100 ps, and the test yield is 63.6%. The above calculation results show that test yield and quality will continue to decline. As a result, even with the continuous advancement of semiconductor manufacturing technology, the test prospects are bleak if no advanced technological breakthrough occurs in future testers. Consequently, we propose a new test scheme (TRTS) that extends the test time and moves the TGB to improve IC tester capability and test yield.

We used the repeated test method to estimate the test yield of chips produced in 2021 after changing the test method under the same ATE equipment (OTA = 100 ps). Repeated testing of the $R_{3t}^{3p}$ method yielded a test yield of 81.7%), which is about 12.3% higher than that obtained by the traditional test method $R_{1t}^{1p}$ (69.4%). Next, we refer to the 2021 IRDS datasheet and estimate chip production in 2025. Test the DUT using the repeated test $R_{3t}^{3p}$ method, the test yield will increase to Y_t_ = 78.3%, and the repeat test method (TRTS) improves about 14.7% (78.3% − 63.6% = 14.7%) test yield. As compared, repeated tests can improve the tester’s performance and significantly improve the test result. The overall revenue and profit of the company will significantly improve using the test guardband to reduce the incidence of killing errors and achieve high-yield delivery.

Retest has been widely used in the IC test industry, effectively improving the test yield. However, endless blind retesting can cause the cost of testing to outweigh the profit from retesting. As a result, we proposed a three-repetition tests scheme to meet consumer demand for the desired product output (TRTS). According to the obtained test yield and test cost, calculate the required number of retests under the feedback of profit calculation. Through repeated test methods and cost calculations, we verify that the triple test solution (TRTS) maximizes test yields and increases company profits. It saves wasted manpower and time and also minimizes testing costs. Through the three-repetition test (TRTS) mechanism, the best balance of increasing profit and test cost is achieved under the premise of improving the test yield.

### 5.1. Method for Testing Near-Zero Defect Chips

Moore’s Law predicts that the chip’s performance will double every 18 months. As the annual improvement of the integrated circuit is 30%, the accuracy of the tester improves very slowly, increasing by only 12% per year [12,13,14]. If the tester’s progress continues to stagnate, Product quality and yield will be put to the test in the future, putting a lot of pressure on manufacturers. At present, the scale of the automotive electronics market is getting larger and larger, and the scope of application is getting wider and wider in several systems such as braking, power, active stability control, and traction control. We all know stable chips must fully control that complex system control. Therefore, zero-defect electronic chips not only master safety but also become the industry’s goal. In order to ensure the reliability of crucial electronic products, strict quality control is required to eliminate all defective parts in the total number of parts. However, the progress of IC tester (ATE) lags behind the progress of semiconductor manufacturing; thus, the progress of semiconductor test equipment has stagnated, and the test yield rate has dropped. Hence, it will be a challenge to use a tester whose performance lags behind the process capability to pick out electronic products with high reliability. Near-zero defect products [28,29,30,31,32] are the constant pursuit of the semiconductor industry, and there are higher requirements for the safety and quality of the biomedical or automotive electronics industries. Semiconductor chips often use defect per million defect rate to indicate quality. Furthermore, suppliers will increase the defect rate to one in a billion (Parts Per Billion, PPB) in some critical parts such as avionics or automobiles, which can reduce the malfunction of electronic parts and improve aviation safety or automobile driving.

Refer to the IRDS 2021 datasheet (Figure 12 and Table 3), limit the DL to 10 ppm, and use the traditional test method $R_{1t}^{1p}$ to test the chips (DUT) produced in 2021. Using the OTA tester (ATE) with 100 ps characteristic parameters to test the DUT, the test specification is set to TS = 203 ps, and we get Y_t_ = 50.5%. Under the same IC tester (OTA = 100 ps) equipment, we changed the test method and used the repeat test method $R_{3t}^{3p}$ to test the chips (DUT) produced in 2021. According to the estimated results, the test yield (71.1%) obtained by the repeated test method $R_{3t}^{3p}$ is about 20.6% higher than that obtained using the traditional test method $R_{1t}^{1p}$ (71.1%).

Next, test chips produced in 2025 with electrical parameters DS = 278 ps (3.60 GHz), using the tester test parameter OTA = 100 ps, the test specification is set to TS = 166 ps, and we can get 42.5% test yield. We changed the test method under the same IC tester equipment (OTA = 100 ps) and tested the DUT using the repeated test method. The test yield (66.1%) obtained by repeated testing of the $R_{3t}^{3p}$ method is about 23.6% higher than that obtained by the traditional test method $R_{1t}^{1p}$ (42.5%). Using repeat testing to select high-quality chips (i.e., DL = 10 ppm) reduces the number of chips that are killing errors and increases the number of high-quality chips that can be sold. Undoubtedly, repeated testing (TRTS) can improve the performance of the semiconductor test equipment, and the test yield can also be significantly improved. Hence, as long as test vendors are willing to provide useful test methods, chips with miskill errors can be removed from the defect pile, but high-quality product delivery can be achieved while increasing the company’s overall profit.

### 5.2. Estimate the Company’s Increased Profits from a Cost-Benefit Perspective

Using the previous example to estimate the additional company profit when using repeated testing from a cost standpoint. For example, suppose company “B” manufactures 100 million chips per year, and the test cost per chip is 5% of the manufacturing cost. In this case, we use the 8:20 international chip pricing strategy. This pricing ratio is used by major manufacturers and can also estimate to forecast the company’s future chip costs and sales profits. Assuming the IC costs $8 to manufacture, the IC sells for $20, the total cost of required testing is about $40 million (100,000,000 × 8 × 5% = $40,000,000). We first consider the general electronic product quality and set the test quality to DL = 300 ppm. Referring to the above example, the repeat test $R_{3t}^{3p}$ and moving the TGB can increase the test yield from 63.6% to 78.3% in 2025 (Figure 12 and Table 3). Subtracting the cost of testing three times, repeating the test can increase the profit by $147 million (100 million × 20 × 14.7% = $294 million, 294 − 40 − 40 − 40 = $174 million).

Following the above, we consider high-quality electronic chips in the biomedical or automotive fields, and we set the test quality at DL = 10 ppm. Using the repeat test $R_{3t}^{3p}$ and moving the TGB can increase the test yield from 42.5% to 66.1% in 2025 (Figure 13). Subtracting the cost of testing three times, repeating the test (TRTS) can increase the profit by $352 million (100 million × 20 × 23.6% = $294 million, 472 − 40 − 40 − 40 = $352 million).

We all know that Zero-Defect Manufacturing (ZDM) [32] is the ultimate goal of all industries. The requirements for the chip quality required for biomedical avionics are quite stringent. In terms of price, the unit price of avionics chips is naturally many times higher than ordinary quality chips. Therefore, we adopt repeated testing methods to reduce errors in the testing process by using a test guardband to minimize killing and omission errors and obtain products with near-zero defects [28,29,30,31,32]. Repeated testing (TRTS) is applied to high-quality chip testing (10 ppm), and the test yield is improved more than the general (300 ppm) test yield. The number of chips that can be sold has also increased a lot (to earn more profits through quality improvement). Consequently, the profit will also increase several times, significantly improving the company’s profits.

We applied the repeated test method (TRTS) in the 2021 IRDS table, which improved the test yield by more than 28% compared with the traditional test method. Retesting is known to be widely used in the IC testing industry to effectively improve test yield. However, endless undirected retesting can cause the cost of testing to outweigh the profit of retesting. Therefore, in order to meet consumers’ requirements for the high-quality products they need, we have proposed a three-repetition tests scheme (TRTS). Under the feedback of cost calculation, the best test specifications and results are calculated based on the obtained test yield and test cost. Through IDRS 2021 data and profit cost calculation, the TRTS approach can maximize company profits and increase annual chip production, reducing the impact of the COVID-19 pandemic on-chip capacity.

### 5.3. The Triple Test Scheme Has Advantages and Future Development Prospects

The size of components and the width of the wires on the chip are gradually shrinking as the semiconductor manufacturing process advances. Considering the actual physical characteristics, the complexity of the chip will double in 18 to 24 months. However, suppliers face the rapid progress of semiconductor manufacturing technology and the slow development of testing technology. Using existing instruments and tools to pick out electronic products with high reliability will be a great challenge. As the quality requirements of current electronic products are stringent, suppliers must re-evaluate their test procedures and methods to find more effective alternative test methods. The triple test scheme has the following advantages and future development prospects.

(1) The three-repetition tests scheme is to change the test specifications(move test guardband) and retest the object to be tested multiple times. Spending more test time can reduce the number of mis-killed chips and bring about a higher test yield. The test yield of the triple test solution is better than the traditional test method. As long as the manufacturer is willing to spend more time on testing, not only chips with fatal errors can be removed from the defect heap, but high-yield deliveries can also be achieved.

(2) Testing and manufacturing technology develop at different rates, and testing technology lags far behind design and manufacturing technology. The current development of the test machine is quite disappointing, and the vision for the future outlook is rather pessimistic due to insufficient test capabilities. We proposed a three-repetition tests scheme, which can improve the ability of the tester to distinguish between Good and Bad products. It has the potential to improve test results and the performance of the IC tester.

(3) The optimized three-repetition tests scheme can avoid blind retesting. It saves wasted manpower and time and also minimizes testing costs. The best balance of increasing profit and test cost is achieved through the test mechanism, which focuses on improving test yield.

(4) Due to the impact of the global pandemic (COVID-19), the production of chips has been greatly reduced. The production capacity of the test plant has dropped, causing many testers to sit idle in the testing house. If an effective test method can be adopted, the idle tester can be fully utilized to improve the test yield and quality. We proposed a three-repetition tests scheme solution system to maximize test yield and increase available chips to solve some of the global chip shortages.

(5) To achieve the goal of close to zero defects, use the moving test guardband strategy to repeatedly search for reliable products close to zero defects. The triple test solution can effectively identify high-quality chips while increasing product shipments. High-quality chips cost several times the price of ordinary chips (high-quality chips, high prices, high profits); thus, the high unit price of the product increases the company’s profit.

## 6. Conclusions

We propose a model for testing quality and yield, which can effectively analyze the impact of the manufacturing process and test parameters on quality and yield. Due to the decline in global semiconductor production capacity caused by the COVID-19 pandemic, test equipment in related semiconductor test factories is idle. In addition to lowering testing costs, it can strictly control the quality of semiconductor chips, resulting in more high-quality products and increased productivity and profits. With changes in consumer consumption patterns and concepts, the demand for zero-defect products is prioritized. However, the development of IC testing technology and semiconductor technology differs, and the testing technology lags far behind the manufacturing technology. Therefore, the ability of the IC tester (ATE) to distinguish high-quality chips is deteriorating. In such a difficult situation, using more effective methods to improve the capabilities of semiconductor test equipment has become a major concern. At present, retesting is used in the production process of the semiconductor test industry to enhance the test yield and test quality. Furthermore, the retest strategy effectively improves test yield and quality in the actual IC test factory. Pursuing high-quality product testing methods, we proposed repeated testing methods (TRTS) to effectively improve post-test yield by moving iterations of test guardband with the poorer performance of IC testers.

We used the DITM model to estimate the future yield trend and applied the re-measurement method using data from the IRDS table in 2021. (TRTS). Retesting can effectively improve test results after obtaining the results of estimation and operation. In general consumer electronics product testing (DL = 300 pm), the number of chips that can be sold has also been increased, boosting the company’s chip sales profits. The relative company profit also grows significantly. The performance is more prominent in testing chips (biomedical electronics, automotive electronics) with strict requirements (DL = 10 ppm). Unnecessary production waste is reduced by minimizing more missing and killing errors. Also, it has improved test yields more than typical consumer electronics products and gained more chip sales. Therefore, facing the shortage of goods and materials in the electronics industry ((COVID-19 impact), the TRTS method can effectively increase the global supply of semiconductor chips. Moreover, more high-quality chips can be picked out, and the profit will naturally grow several times (High-quality chips, high price, and high profit). Hence, the retest strategy (TRTS) not only improves the performance of semiconductor test equipment and reduces test costs but also maximizes company profits.

## Figures and Tables

**Figure 1 sensors-22-04158-f001:**
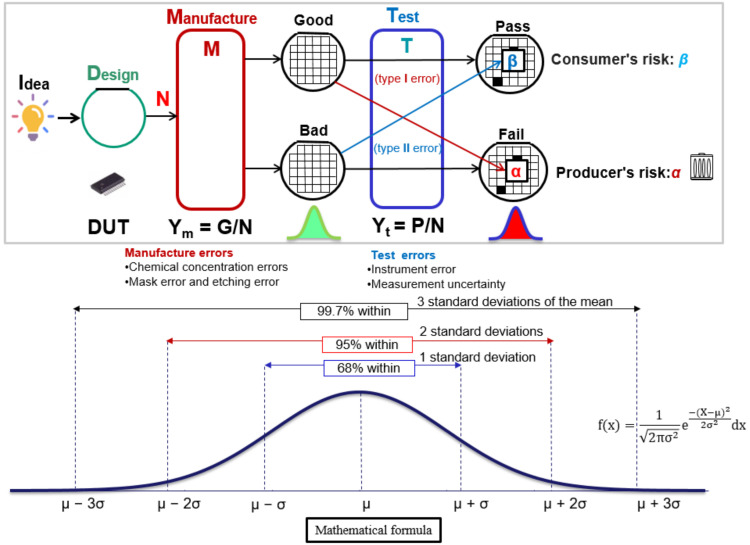
The process of chip fabrication and testing.

**Figure 2 sensors-22-04158-f002:**
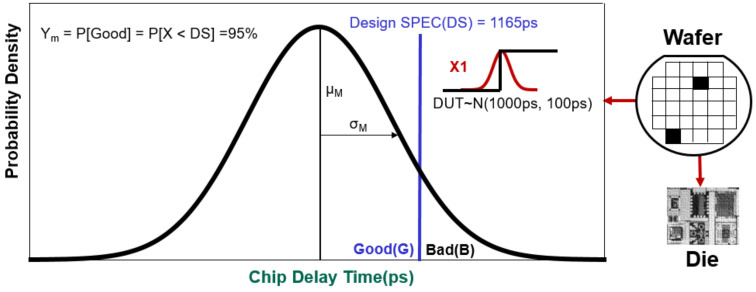
Distribution and calculation of manufacturing yield.

**Figure 3 sensors-22-04158-f003:**
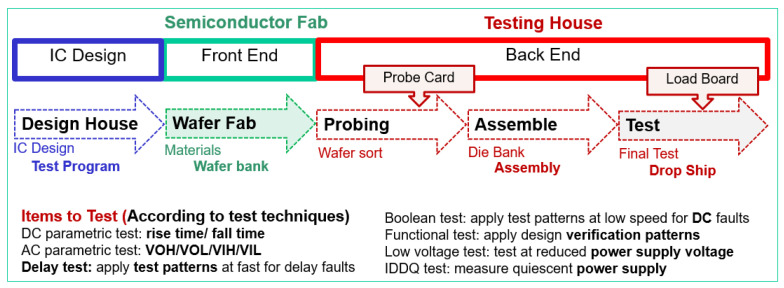
Schematic diagram of the chip development and testing process.

**Figure 4 sensors-22-04158-f004:**
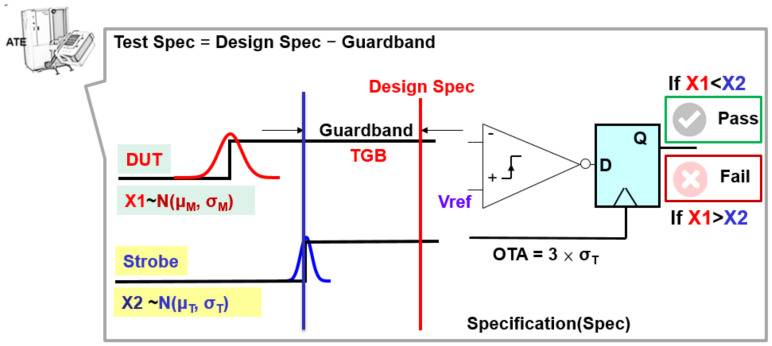
Equivalent model of threshold test tester system.

**Figure 5 sensors-22-04158-f005:**
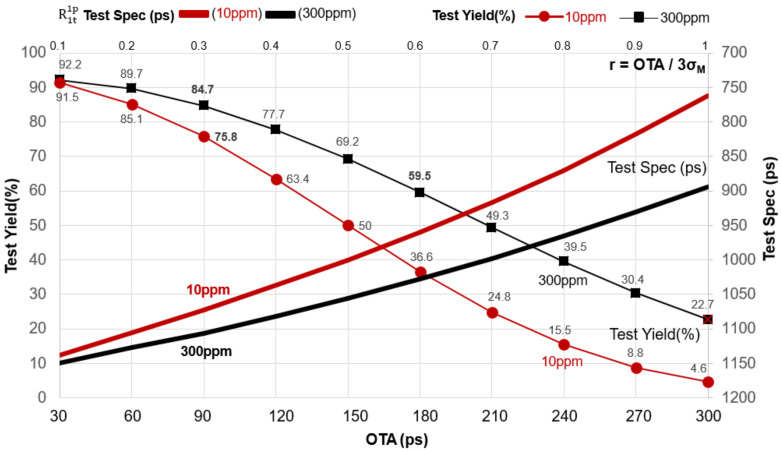
High-precision ATE can increase test yield and chips shipments.

**Figure 6 sensors-22-04158-f006:**
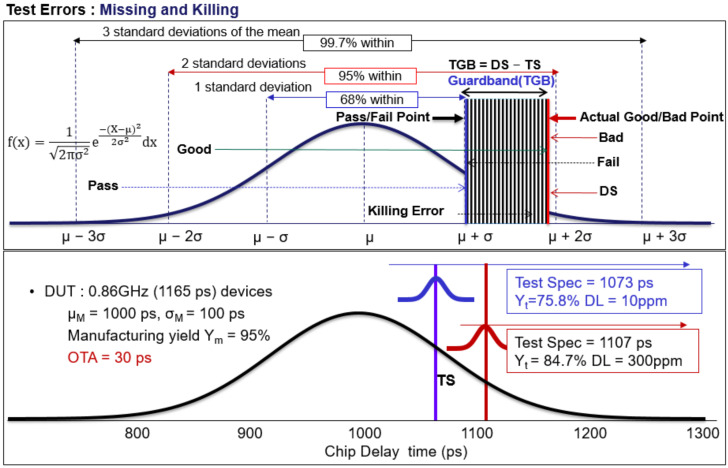
The test guardband affects the test yield and test quality of the test.

**Figure 7 sensors-22-04158-f007:**
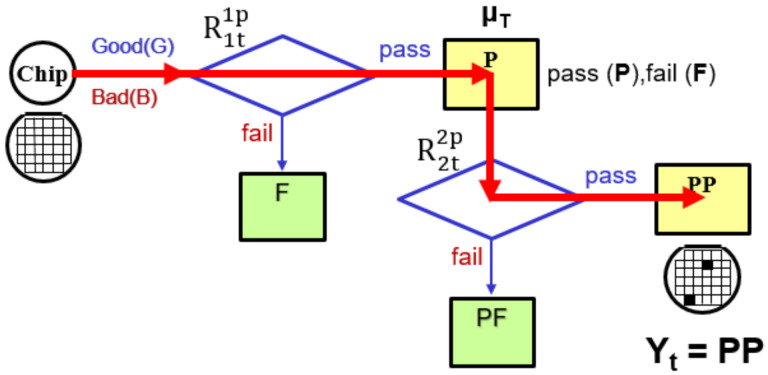
Schematic diagram of the repeated testing ($R_{2t}^{2p}$).

**Figure 8 sensors-22-04158-f008:**
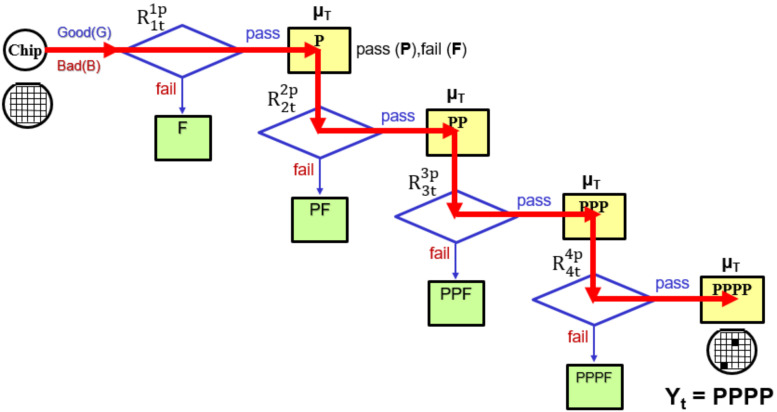
Method of repetition tests scheme $R_{\mathrm{nt}}^{\mathrm{np}}$ to improve test yield.

**Figure 9 sensors-22-04158-f009:**
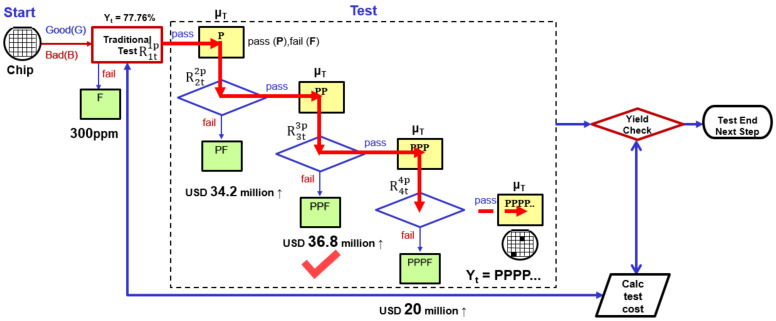
Decision-making mechanisms of the three-repetition tests scheme (300 ppm).

**Figure 10 sensors-22-04158-f010:**
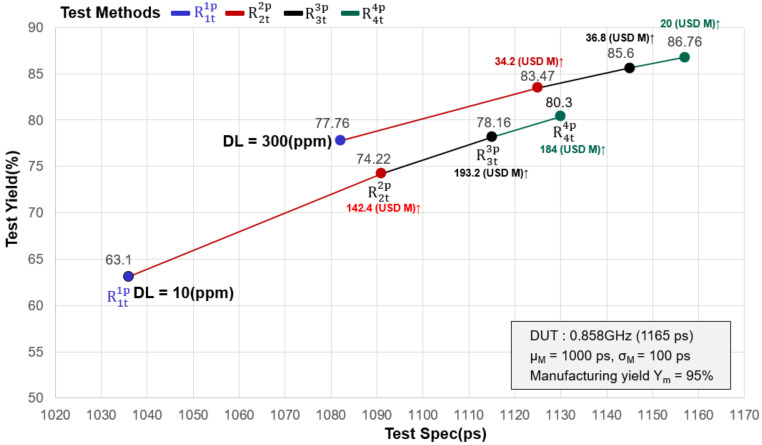
Comparison of the improved test yield of repeated tests (TRTS) under different quality conditions.

**Figure 11 sensors-22-04158-f011:**
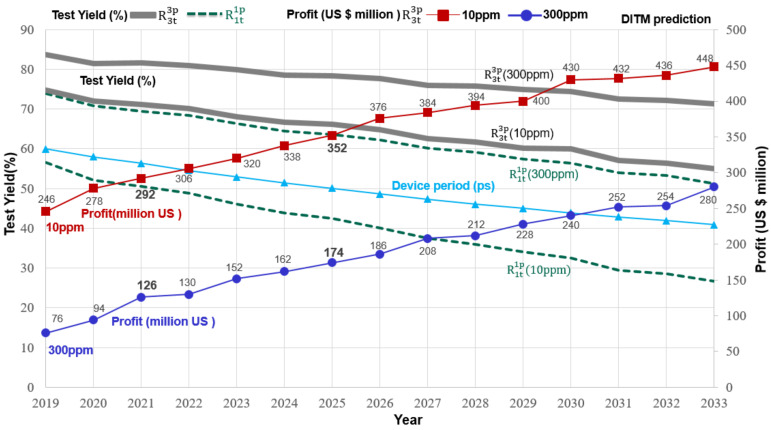
Three-repetition tests scheme (TRTS) improves test yield and overall company profit.

**Figure 12 sensors-22-04158-f012:**
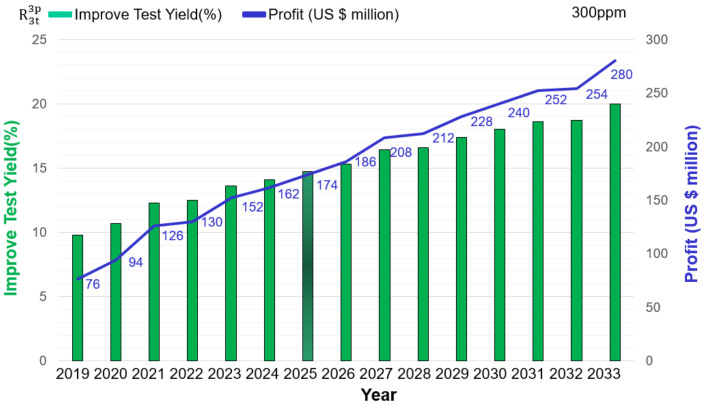
The test method affects the company’s total profit (300 ppm).

**Figure 13 sensors-22-04158-f013:**
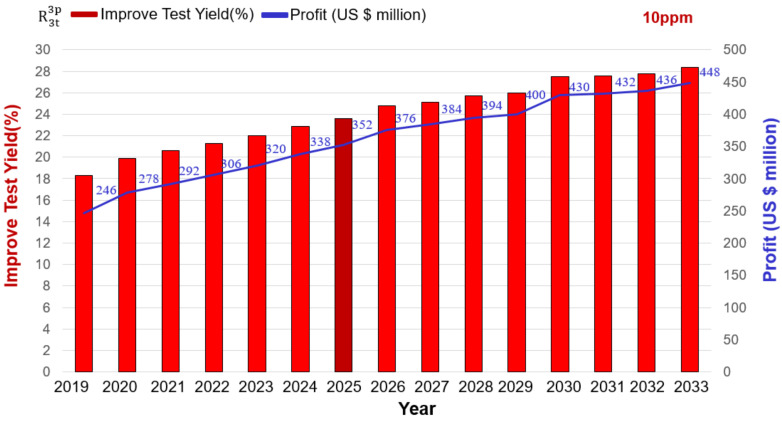
High-quality products increase the company’s total profit (10 ppm).

**Table 1 sensors-22-04158-t001:** Test accuracy impacts test yield and test quality.

**Chip Frequency**	**GHz**	**0.858**	**0.858**	**0.858**	**0.858**	**0.858**	**0.858**	**0.858**	**0.858**	**0.858**	**0.858**
Device period	ps	1165	1165	1165	1165	1165	1165	1165	1165	1165	1165
μ_M_	ps	1000	1000	1000	1000	1000	1000	1000	1000	1000	1000
σ_M_	ps	100	100	100	100	100	100	100	100	100	100
OTA = 3σ_T_	ps	300	270	240	210	180	150	120	90	60	30
σ_T_	ps	100	90	80	70	60	50	40	30	20	10
r = OTA/3σ_M_		1	0.9	0.8	0.7	0.6	0.5	0.4	0.3	0.2	0.1
Test method		$R_{1t}^{1p}$	$R_{1t}^{1p}$	$R_{1t}^{1p}$	$R_{1t}^{1p}$	$R_{1t}^{1p}$	$R_{1t}^{1p}$	$R_{1t}^{1p}$	$R_{1t}^{1p}$	$R_{1t}^{1p}$	$R_{1t}^{1p}$
Y_m_	%	95	95	95	95	95	95	95	95	95	95
DL	ppm	300	300	300	300	300	300	300	300	300	300
TS(μ_T_)	ps	894	931	966	998	1028	1056	1082	1107	1127	1150
Y_t_	%	22.7	30.4	39.5	49.3	59.5	69.2	77.7	84.7	89.7	92.2
DL	ppm	10	10	10	10	10	10	10	10	10	10
TS(μ_T_)	ps	762	818	870	917	960	1000	1037	1073	1106	1138
Y_t_	%	4.6	8.8	15.5	24.8	36.6	50	63.4	75.8	85.1	91.5

**Table 2 sensors-22-04158-t002:** The results of the test depend on the different test methods and the number of tests.

**Chip Frequency**	**GHz**	**0.858**	**0.858**	**0.858**	**0.858**	**0.858**	**0.858**	**0.858**	**0.858**
Device period	ps	1165	1165	1165	1165	1165	1165	1165	1165
μ_M_	ps	1000	1000	1000	1000	1000	1000	1000	1000
σ_M_	ps	100	100	100	100	100	100	100	100
σ_T_	ps	40	40	40	40	40	40	40	40
OTA = 3σ_T_	ps	120	120	120	120	120	120	120	120
r = OTA/3σ_M_		0.4	0.4	0.4	0.4	0.4	0.4	0.4	0.4
Test method		$R_{1t}^{1p}$	$R_{2t}^{2p}$	$R_{3t}^{3p}$	$R_{4t}^{4p}$	$R_{1t}^{1p}$	$R_{2t}^{2p}$	$R_{3t}^{3p}$	$R_{4t}^{4p}$
DL	ppm	300	300	300	300	10	10	10	10
TS(μ_T_)	ps	77.76	83.47	85.60	86.76	63.10	74.22	78.16	80.3
Y_t_	%	1082	1125	1145	1157	1036	1091	1115	1130
Improve Y_t_	%		5.71	7.84	9		11.12	15.66	17.2
Increased maximum profit	million US		34.2↑	36.8↑	20↑		142.4↑	193.2↑	184↑

**Table 3 sensors-22-04158-t003:** Efficient test method for high-quality chips.

Year		Unit	2019	2020	2021	2022	2023	2024	2025	2026	2027	2028	2029	2030	2031	2032	2033
Chip frequency		GHz	3.00	3.10	3.20	3.30	3.40	3.50	3.60	3.70	3.80	3.90	4.00	4.10	4.20	4.30	4.40
Device period		ps	333	322	313	303	294	286	278	270	263	256	250	244	238	233	227
μ_M_		ps	212	208	202	195	190	185	179	174	170	165	161	157	154	150	146
σ_M_		ps	72	69	67	65	63	62	60	58	57	55	54	52	51	50	49
OTA		ps	100	100	100	100	100	100	100	100	100	100	100	100	100	100	100
DL		ppm	300	300	300	300	300	300	300	300	300	300	300	300	300	300	300
$R_{1t}^{1p}$	Y_t_	%	73.9	70.8	69.4	68.4	66.3	64.4	63.6	62.3	60.2	59.2	57.5	56.4	53.9	53.3	51.3
	TS(μ_T_)	ps	263	250	240	230	220	211	203	195	187	180	173	167	160	155	148
$R_{3t}^{3p}$	Y_t_	%	83.7	81.5	81.7	80.9	79.9	78.5	78.3	77.6	76	75.8	74.9	74.4	72.5	72.2	71.3
	TS(μ_T_)	ps	315	303	295	284	275	266	258	250	242	235	229	223	216	211	205
Improve Y_t_		%	9.8	10.7	12.3	12.5	13.6	14.1	14.7	15.3	16.4	16.6	17.4	18	18.6	18.7	20
Increased maximum profit		million US	76	94	126	130	152	162	174	186	208	212	228	240	252	254	280
DL		ppm	10	10	10	10	10	10	10	10	10	10	10	10	10	10	10
$R_{1t}^{1p}$	Y_t_	%	56.5	52.1	50.5	48.9	46.1	43.8	42.5	40	37.5	36	34.1	32.5	29.4	28.6	26.6
	TS(μ_T_)	ps	225	212	203	193	183	174	166	157	149	142	135	129	121	116	109
$R_{3t}^{3p}$	Y_t_	%	74.8	72	71.1	70.2	68.1	66.7	66.1	64.8	62.6	61.7	60.1	60	57	56.4	55
	TS(μ_T_)	ps	291	279	270	260	250	242	234	226	218	211	205	199	192	187	181
Improve Y_t_		%	18.3	19.9	20.6	21.3	22	22.9	23.6	24.8	25.1	25.7	26	27.5	27.6	27.8	28.4
Increased maximum profit		million US	246	278	292	306	320	338	352	376	384	394	400	430	432	436	448

## Data Availability

All data are included within manuscript.

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
