# Peer review of "Application of Three-Repetition Tests Scheme to Improve Integrated Circuits Test Quality to Near-Zero Defect"

_sensors, 2022, doi:10.3390/s22114158_

Round 1

Author Response

Dear Sir,

Please find, in the submission section of the authors, our final response to the comments received from the three reviewers to

" Application of Three-Repetition Tests Scheme to Improve Integrated Circuits Test Quality to Near-Zero Defect."

We appreciate your comments very much, as they have pointed out a number of issues that need to be addressed. We would like to thank the editor and the reviewers for taking time reading and suggesting modifications to the paper, and your cogent comments have proven to be very useful for the improvement of the paper. We did several modifications to the initial manuscript based on the suggestions of the reviewers. We hope that the editor will find the paper eligible for publication. In the answers we have explicate all the changes we have done. We hope that this will be useful for the new review. Thank you very much for your kind consideration of this resubmitted version of our manuscript.

Sincerely yours,

Yeh C-H,

(On behalf of the authors of the manuscript)

Reviewer 2 Report

The manuscript entitled "application of three-repetition tests scheme to improve integrated circuits test quality to near-zero defect" by Yeh and Chen reports a model for testing quality and yield. In fact, three-repetition tests scheme (TRTS) is presented to ensure product quality and to maximize company profits and to increase annual chip production (considering the impact of the COVID-19 pandemic). The reviewer feels that the topic is fair but this subject is not suitable for Sensors. There is no efficient algorithm(s) and the literature survey on the topic is not complete. Please clarify if the three-repetition tests scheme is optimized. Proper references should be considered for the equations. Also, the novelty should be highlighted and the results needs to be verified.

Author Response

(The authors gave the same response as above.)

Reviewer 3 Report

First of all, thank you very much for the material sent.

Nevertheless, I have some suggestions:

  1. I am asking for a much better, more transparent description of the current methods of testing and quality control of electronic components in the world. Otherwise, one can be convinced that nothing is happening on this topic at present, and the devices sold, for example, by Intel, are incorrectly verified and hence the company does not earn.
  2. How many physical devices have been validated and to what extent? This is not clearly seen.
  3. What are the development prospects of the proposed method? This is important due to the rapidly changing production, the authors point out.
  4. There are very few teams in the literature review. Aren't others in the world doing this? There are bands from Israel, USA, Canada, Switzerland, Germany and others.

Author Response

(The authors gave the same response as above.)

Round 2

Reviewer 2 Report

The revised manuscript can be accepted for publication in Sensors, however, it needs to be carefully edited before the publication. Congratulation!

Reviewer 3 Report

Thank you very much. I accept the job and congratulations. Best regards